# Expression of Translocator Protein and [18F]-GE180 Ligand Uptake in Multiple Sclerosis Animal Models

**DOI:** 10.3390/cells8020094

**Published:** 2019-01-28

**Authors:** Anne Nack, Matthias Brendel, Julia Nedelcu, Markus Daerr, Stella Nyamoya, Cordian Beyer, Carola Focke, Maximilian Deussing, Chloé Hoornaert, Peter Ponsaerts, Christoph Schmitz, Peter Bartenstein, Axel Rominger, Markus Kipp

**Affiliations:** 1Department of Anatomy II, Ludwig-Maximilians-University of Munich, 80336 Munich, Germany; A.Nack@campus.lmu.de (A.N.); julia.nedelcu@campus.lmu.de (J.N.); Markus.Daerr@campus.lmu.de (M.D.); stella.nyamoya@rwth-aachen.de (S.N.); Christoph_Schmitz@med.uni-muenchen.de (C.S.); 2Department of Nuclear Medicine, University Hospital, LMU Munich, 80336 Munich, Germany; Matthias.Brendel@med.uni-muenchen.de (M.B.); Carola.Focke@med.uni-muenchen.de (C.F.); Maximilian.Deussing@med.uni-muenchen.de (M.D.); Peter.Bartenstein@med.uni-muenchen.de (P.B.); Axel.Rominger@insel.ch (A.R.); 3Institute of Neuroanatomy, RWTH Aachen University, 52074 Aachen, Germany; CBeyer@ukaachen.de (C.B.); 4Laboratory of Experimental Hematology, University of Antwerp, 2000 Antwerp, Belgium; chloe.hoornaert@uantwerpen.be (C.H.); peter.ponsaerts@uantwerpen.be (P.P.); 5Vaccine and Infectious Disease Institute (Vaxinfectio), University of Antwerp, 2000 Antwerp, Belgium; 6Department of Nuclear Medicine, Inselspital, University Hospital Bern, 3010 Bern, Switzerland; 7Department of Anatomy, Rostock University Medical Center, 18055 Rostock, Germany

**Keywords:** multiple sclerosis, cuprizone, TSPO, PET, neurodegeneration

## Abstract

Positron emission tomography (PET) ligands targeting the translocator protein (TSPO) represent promising tools to visualize neuroinflammation in multiple sclerosis (MS). Although it is known that TSPO is expressed in the outer mitochondria membrane, its cellular localization in the central nervous system under physiological and pathological conditions is not entirely clear. The purpose of this study was to assess the feasibility of utilizing PET imaging with the TSPO tracer, [18F]-GE180, to detect histopathological changes during experimental demyelination, and to determine which cell types express TSPO. C57BL/6 mice were fed with cuprizone for up to 5 weeks to induce demyelination. Groups of mice were investigated by [18F]-GE180 PET imaging at week 5. Recruitment of peripheral immune cells was triggered by combining cuprizone intoxication with MOG_35–55_ immunization (i.e., Cup/EAE). Immunofluorescence double-labelling and transgene mice were used to determine which cell types express TSPO. [18F]-GE180-PET reliably detected the cuprizone-induced pathology in various white and grey matter regions, including the corpus callosum, cortex, hippocampus, thalamus and caudoputamen. Cuprizone-induced demyelination was paralleled by an increase in TSPO expression, glia activation and axonal injury. Most of the microglia and around one-third of the astrocytes expressed TSPO. TSPO expression induction was more severe in the white matter corpus callosum compared to the grey matter cortex. Although mitochondria accumulate at sites of focal axonal injury, these mitochondria do not express TSPO. In Cup/EAE mice, both microglia and recruited monocytes contribute to the TSPO expressing cell populations. These findings support the notion that TSPO is a valuable marker for the in vivo visualization and quantification of neuropathological changes in the MS brain. The pathological substrate of an increase in TSPO-ligand binding might be diverse including microglia activation, peripheral monocyte recruitment, or astrocytosis, but not axonal injury.

## 1. Introduction

Multiple sclerosis (MS), an inflammatory disorder of the central nervous system (CNS), is associated with the development of demyelinated plaques, oligodendrocyte destruction and axonal degeneration. These pathological processes are paralleled by the activation of astrocytes and microglia, as well as the recruitment of peripheral immune cells into the central nervous system (CNS) [1,2]. Notwithstanding the traditional focus on white matter tracts as the predominantly affected regions in the CNS of MS patients, demyelination also occurs in various grey matter brain regions [3,4,5], including the hippocampus [6], neocortex [7], thalamus [5], and caudoputamen [5,8]. MS can present in different clinical courses. Relapsing-remitting MS (RRMS) is characterized by single attacks (also known as relapses) with full or partial recovery. There is minimal to no disease progression during the periods between two relapses. Secondary progressive MS (SPMS) is characterized by an initial RRMS disease course, followed by gradual worsening with or without occasional relapses. During this progressive disease course, progression occurs independent of the relapses. Primary progressive MS (PPMS) is characterized by the progressive accumulation of disability from disease onset. During the RRMS disease course, multifocal inflammatory lesions dominate the pathological process, whereas the gradual accumulation of irreversible disability that characterizes progressive MS seems to result more from diffuse immune mechanisms and neurodegeneration [1,9]. As a result, the currently approved drugs for the treatment of RRMS have marginal or no efficacy in progressive MS without inflammatory lesion activity. The development of effective therapies for progressive MS that prevent worsening, reverse damage, and restore function are a major unmet medical need [9].

The pathogenic cascade(s) underlying neurodegeneration is incompletely understood. Recent data suggest that CNS-resident microglia as well as infiltrating inflammatory monocytes play a key role in neurodegeneration [10,11]. For example, microglia cells together with newly recruited blood-borne macrophages orchestrate a new demyelination process by myelin phagocytosis [12]. Moreover, microglial cells are considered to be the main antigen presenting cell population in the CNS and might therefore play important roles during lymphocyte reactivation [13,14]. In the brains of MS patients, microglial activation is not restricted to the focal inflammatory lesions but also diffusely present in normal appearing white and gray matter areas. Therefore, diffuse microglia activation might be responsible for the diffuse neurodegeneration seen in MS patients [15]. Furthermore, in preclinical MS models acute axonal injury predominantly occurs in regions with pronounced microglia activation [16]. If microglia sense injury, they are rapidly activated and respond by the production of cytokines and chemokines, increased phagocytosis, the release of reactive oxygen species as well as by antigen presentation [17,18]. By these mechanisms, microglia can trigger the activation of neurotoxic pathways leading to progressive neurodegeneration. Manipulation of microglia cell function might, therefore, be a valuable strategy to ameliorate or even prevent neurodegeneration in MS patients, and in consequence alleviate disease progression. However, formal proof that the (prolonged) presence of activated microglia is indeed detrimental in MS remains to be established. Hence, a valid method to measure microglia activation in MS patients is urgently needed.

One of the molecules that is highly expressed by activated microglia is the mitochondrial translocator protein 18 kDa (TSPO), formerly known as the “peripheral benzodiazepine receptor” [19]. TSPO is a receptor that is part of a multimeric complex. This complex includes a voltage-dependent anion channel and an adenine nucleotide carrier and locates to the outer mitochondrial membrane [20]. There, TSPO is believed to play crucial roles in cell physiology and development, as suggested by its sequence conservation from bacteria to humans [21]. Although TSPO is believed to be important for maintaining the mitochondrial membrane potential, this protein is also critically involved in cholesterol transport, making it essential for steroidogenesis [22]. In addition, TSPO plays roles in cellular proliferation, apoptosis and inflammation as well as in porphyrin transport and haem biosynthesis (see [23,24]) for excellent reviews]. In the CNS, TSPO is thought to be mainly expressed by activated microglia cells, although various other cell types, including endothelial and smooth muscle cells, astrocytes and ependymal cells, were shown to express TSPO [25]. In MS tissues and other CNS disorders, microglia and macrophages are believed to represent the bulk of TSPO^+^ cells [25,26,27]. However, astrocytes have also been shown to upregulate TSPO expression during CNS injuries [26,28]. In fact, the relative contribution of astrocytes to the global binding of TSPO ligands has been, and still is, a matter of debate [29].

Today, conventional MR imaging (MRI) techniques represent an important step during MS diagnostics and clinical follow-up. It is sensitive in demonstrating focal inflammatory lesions and, to a certain extent, diffuse brain atrophy. However, the widespread diffuse pathology within the normal appearing white matter as well as grey matter pathology, which is often related to microglial activation and neurodegeneration, cannot be reliably detected using conventional MRI [30]. Positron emission tomography (PET) is a sensitive method for visualizing MS-related processes at the molecular level, and is a useful tool to visualize neuroinflammation and the activation of microglia [31]. In contrast to other imaging modalities, PET can visualize cell metabolism in real time, and physiological parameters can be quantified in active disease processes. TSPO is the most widely used target in PET imaging of neuroinflammation. The first successful and now most frequently used TSPO-radioligand is (11C) PK11195 [32]. However, (11C) PK11195 has several disadvantages, including limited brain entrance, a poor signal-to-noise ratio and labelling with the impractically rapid decaying isotope, (11C) [31]. This led to the development of a wide range of second- and third-generation TSPO radioligands, including (18F)-GE180 [33] which indicated good performance in MS patients [34,35].

The cellular source of TSPO expression, and hence TSPO-ligand binding in the CNS of MS patients is not entirely clear. While most studies suggest that TSPO is predominatly expressed by activated microglia cells and in consequence an increase in TSPO ligand binding is interpretated as microglia activation [36,37,38,39,40], various other cell types have been shown to express TSPO as well, including astrocytes [41,42]. The aim of this study was, therefore, to establish the applicability of the third-generation TSPO ligand, (18F)-GE180, to (semi-)quantify glia cell activation and neuroinflammation in a MS mouse model, and to directly correlate ligand binding with TSPO expression. Furthermore, we readdressed which cell types express TSPO after experimental demyelination with a focus on astrocytes, microglia cells and sites of acute axonal injury.

## 2. Material and Methods

### 2.1. Animals and Experimental Groups

All in vivo experiments were performed as published previously with minor modifications [43,44,45]. Eight-week-old C57BL/6 female mice (19–20 g) were purchased from Janvier Labs, Le Genest-Saint-Isle, France. The mice were allowed to accommodate to the environment for at least 1 week prior to the beginning of the experiment. The hGFAP/EGFP transgenic mice [46] were used to visualize entire astrocyte cell bodies and processes. The CX_3_CR1^+/eGFP^/CCR2^+/RFP^-mice with RFP-expressing monocyte-derived macrophages and eGFP-expressing microglia [47] were used to analyze whether TSPO is expressed in microglia and/ or monocytes. Microbiological monitoring was performed according to the Federation of European Laboratory Animal Science Associations recommendations. A maximum of five animals were housed per cage (cage area 435 cm^2^). The animals were kept at a 13 h light/11 h dark cycle, with a controlled temperature of 22 ± 2 °C and 50 ± 10% humidity, with access to food and water ad libitum. It was assured that no light was used during the night cycle period. Nestlets were used for environmental enrichment. The housing conditions for the PET scanned mice were slightly different, e.g., 12 h light/ 12 h dark cycle and six animals per cage. All experiments were formally approved by the *Regierung Oberbayern* (reference number 55.2-154-2532-73-15). The mice were randomly assigned to the following experimental groups: (A) control (co), the animals were provided a diet of standard rodent chow for the entire duration of the study; (B) cuprizone, the animals were intoxicated with a diet containing 0.25% cuprizone (bis(cyclohexanone)oxaldihydrazone; Sigma-Aldrich, Taufkirchen, Germany) mixed into ground standard rodent chow for one week (1 wk cup), three weeks (3 wks cup), or five weeks (5 wks cup); (C) Cup/EAE, the mice were intoxicated with the cuprizone diet for the first three weeks, and were then immunized with MOG_35–55_ at the beginning of week six as published previously [43,44]; (D) EAE, the animals received the standard rodent chow for the duration of the study and were immunized with MOG_35–55_ at the beginning of week six.

### 2.2. EAE and Disease Scoring

EAE scoring was daily performed as published previously [43]. To induce the formation of encephalitogenic T cells, the mice were immunized (s.c.) with an emulsion of MOG_35–55_ peptide dissolved in complete Freund’s adjuvant followed by injections of pertussis toxin in PBS (i.p.) on the day of and the day after immunization (Hooke Laboratories, Inc., Lawrence, USA). The disease severity was scored as follows: A score of 1 was assigned if the entire tail droped over the finger of the observer when the mouse was picked up by the base of the tail; a score of 2 was assigned when the legs of the mice were not spread apart but held close together when the mouse was picked up by the base of the tail, or when mice exhibited a clearly apparent wobbly gait; a score of 3 was assigned when the tail was limp and the mice showed complete paralysis of hind legs (a score of 3.5 is given if the mouse is unable to raise itself when placed on its side); a score of 4 was assigned if the tail was limp and the mice showed complete hind leg and partial front leg paralysis, and the mouse was minimally moving around the cage but appears alert and feeding. A score of 4 was not attained by any of the mice in our study.

### 2.3. Positron Emission Tomography (PET)—Imaging

All rodent PET procedures followed an established standardized protocol for radiochemistry, acquisition and post-processing [48,49]. In brief, [18F]-GE180 TSPO-PET (10.6 ± 2.1 MBq) with an emission window of 60–90 min p.i. was used to measure cerebral microglial activity by a Siemens Inveon DPET (Siemens, Knoxville, Tennessee). All analyses were performed using PMOD (V3.5, PMOD technologies, Basel, Switzerland). Normalization of the injected activity was performed by the previously validated myocardium correction method [50]. TSPO-PET values, derived from a predefined VOI (volume of interest) set (medial corpus callosum (2.2 mm^3^), the lateral corpus callosum (2.9 mm^3^), caudoputamen (4.4 mm^3^), thalamus (4.4 mm^3^), hippocampus (4.4 mm^3^), and cortex (6.7 mm^3^)) were extracted and compared between groups (co versus 5 wks cup) by a Mann–Whitney test. A visual comparison was performed by the calculation of a standardized Z-score difference map for each cerebral voxel for control versus 5 wks cup. To this end, the average and standard deviation maps were generated for the groups of control and 5 wks cup, and the difference between the control and 5 wks cup average maps was scaled by the standard deviation map of control.

### 2.4. Gene Expression Studies

The gene expression levels were semi-quantified by real-time reverse transcription-PCR ([qRT-PCR] Bio-Rad, Munich, Germany), using SensiMix Plus SYBR and fluorescein (Quantace, Bioline, Luckenwalde, Germany) with a standardized protocol as described previously by our group [51]. The primer sequences and individual annealing temperatures are shown in Table 1. Relative quantification was performed using the ΔΔCt method. The β-actin expression levels were used as the reference. Gel electrophoresis and melting curves of the PCR products were routinely performed to determine the specificity of the PCR reactions (data not shown). To exclude contamination of the reagents with either RNA or DNA, appropriate negative controls were performed.

### 2.5. Tissue Preparation

For the histological and immunohistochemical studies, the mice were anaesthetized with ketamine (100 mg·kg^−1^ i.p.) and xylazine (10 mg·kg^−1^ i.p.), and then transcardially perfused with ice-cold PBS followed by a 3.7% formaldehyde solution (pH 7.4). The brains were postfixed in a 3.7% formaldehyde solution overnight, dissected, embedded in paraffin, and then the coronal sections (5 μm) were prepared. For the gene expression studies, after transcardial PBS perfusion, the tissues were manually dissected and cut with a vibratom into 400 µm slices. The hippocampus, corpus callosum and cortex tissues were dissected, and the samples were stored at 0 °C in homogenization tubes filled with 1 mL peqGold TriFast (Peqlab Biotechnologie GmbH, Erlangen, Germany) until further processing.

### 2.6. Immunohistochemistry

Immunohistochemistry was performed as previously published by our group [43,45]. In brief, the antigens were unmasked with heating in either Tris/EDTA (pH 9.0) or citrate (pH 6.0) buffer if appropriate. After washing in PBS, the sections were blocked in blocking solution (serum of the species in which the secondary antibody was produced) for 1 h. Then, the sections were incubated overnight (4 °C) with the primary antibodies diluted in blocking solution. The primary antibodies used in this study are given in Table 2. The next day, the slides were treated with 0.3% hydrogen peroxide in PBS for 30 min and then incubated with biotinylated secondary antibodies for 1 h followed by peroxidase-coupled avidin-biotin complex (ABC kit; Vector Laboratories, Peterborough, UK). The secondary antibodies used in this study are listed in Table 3. The sections were finally exposed to 3,3’-diaminobenzidine (DAKO, Santa Clara, CA, USA) as a peroxidase substrate as published previously [16]. The negative control sections, without primary antibodies or with isotype antibodies, were processed in parallel to ensure specificity of the staining. The stained and processed sections were digitalized using a Nikon ECLIPSE 50i microscope (Nikon, Nikon Instruments, Düsseldorf, Germany) equipped with a DS-2Mv camera. The open source program, ImageJ (NIH, Bethesda, MD, USA), was used to determine the relative area stained by the anti-TSPO antibodies.

### 2.7. Immunofluorescence Labeling

For immunofluorescence stains, the sections were rehydrated, unmasked, and blocked in the serum of the species in which the secondary antibodies were raised. The sections were incubated overnight (at 4 °C) with the indicated combination of primary antibodies diluted in blocking solution. The primary antibodies used in this study are listed in Table 2. For double-labelling experiments, the anti-TSPO antibodies were either combined with anti-IBA1 for the detection of microglia/macrophages, or anti-GFAP for the detection of astrocytes. Acute axonal injury was visualized with anti-APP antibodies. After extensive washing, the sections were incubated with a combination of fluorescent secondary antibodies (Table 3). For the staining of cell nuclei, the sections were then incubated with Hoechst 33258 (bisBenzimide H 33258 Sigma Aldrich, Steinheim, Germany; 1:10,000) diluted in PBS. In parallel, the negative controls were performed by first incubating sections with the primary antibodies and subsequently incubating these sections with the “wrong” secondary antibodies to exclude unspecific binding of the fluorescent secondary antibodies to primary antibodies [52]. By incubating the sections with each of the fluorescent secondary antibodies alone, unspecific secondary antibody binding to the tissue itself was excluded. The stained and processed sections were documented using an Olympus BX41-Wi fluorescence microscope station (Olympus, Hamburg, Germany). The software, Stereo Investigator (MBF Bioscience, Williston, ND, USA), was used to determine the number of single and double-positive cells/spheroids.

### 2.8. Ultrastructural Analysis via Serial Block-Face Scanning Electron Microscopy

Myelinated and demyelinated corpora callosa (i.e., 3 wks cuprizone) were analyzed by 3D EM as published by Ohno and colleagues [53]. To this end, the mice were perfused via the left ventricle with 2.5% (wt/vol) glutaraldehyde and 4% paraformaldehyde. The brains were subsequently removed, stained with heavy metals, and embedded in resin. 3D-electron microscopy (Serial block-face scanning EM) was performed by using a SigmaVP scanning electron microscope (Carl Zeiss) equipped with a 3View in-chamber ultramicrotome system (Gatan). Serial image sequences were generated at 80 nm-nm steps, providing 204.54 μm × 61.36 μm-wide image stacks at a resolution of 0.001 µm per pixel. The images were processed and measured with the open source program, Reconstruct (BU, Boston, MA, USA). The focus of this analysis was to detect mitochondria within axonal swellings.

### 2.9. Statistical Analysis

All data are given as arithmetic means ± SEMs. A *p*-value of <0.05 was considered to be statistically significant. Statistical analyses were performed using Prism 5 (GraphPad Software Inc., San Diego, CA, USA). Applied statistical tests are given in the respective figure legends.

## 3. Results

### 3.1. Cuprizone Intoxication Induces Reproducible Demyelination and Glia Activation

First, we analyzed the spatial distribution of cuprizone-induced demyelination and glia activation after a 5-week cuprizone intoxication protocol. Compared to the control animals, severe loss of anti-PLP staining intensity was evident in several brain regions, including the corpus callosum, grey matter cortex and caudoputamen (see Figure 1A). At the level of the anterior commissure, demyelination of the corpus callosum was prominent at the lateral parts, whereas, at the level of the rostral hippocampus, the medial corpus callosum was most severely demyelinated. Moreover, moderate but consistent demyelination was observed in the hippocampus. To analyze the extent of microglia and astrocyte activation, anti-IBA1 and anti-GFAP immunohistochemical stains were performed, respectively. As shown in Figure 1B, C, demyelination was paralleled by severe microgliosis and astrocytosis. Anti-IBA1 and anti-GFAP staining intensity was most severe within the demyelinated corpus callosum, less so in the affected grey matter cortex and hippocampus. Within the caudoputamen, microgliosis and astrocytosis were severe within the demyelinated striosomes, and moderate in the matrix region.

### 3.2. TSPO Expression and (18F)-GE180 Uptake Correlate with the Degree of Demyelination and Glial Cell Activation

Different cell types have been discussed to induce TSPO expression in response to brain injuries, including microglia and astrocytes. During inflammatory demyelination, a third cell type might contribute to the global brain TSPO load, namely, recruited monocytes. In a first step, we applied the reductionistic cuprizone demyelination model where demyelination occurs despite the presence of an intact blood–brain barrier, and, in consequence, the absence of peripheral monocytes and lymphocytes [54]. In that model, we performed μPET analyses in control and 5-week cuprizone-treated animals to quantify uptake of the novel TSPO ligand, (18F)-GE180 [55], and compared ligand uptake to histopathological changes.

In line with previous results [41,56], we found a region-dependent increase of [18F]-GE180 uptake in cuprizone-treated mice (Figure 2A). A visual interpretation of the TSPO μPET scans indicated increased tracer uptake, particularly in the corpus callosum, caudoputamen, thalamus and cortex regions (Figure 2B). Statistical comparison between the control and cuprizone-intoxicated mice showed significant higher tracer uptake in the medial corpus callosum (∆36 ± 4%, *p* < 0.001), the lateral corpus callosum (∆58 ± 5%, *p* < 0.001), the caudoputamen (∆65 ± 5%, *p* < 0.001), the thalamus (∆47 ± 5%, *p* < 0.001), the hippocampus (∆46 ± 5%, *p* < 0.001), and the cortex (∆54 ± 5%, *p* < 0.001).

In order to relate the observed increased tracer uptake in vivo to the histological changes, we analyzed TSPO reactivity in the different brain regions. In control animals, the anti-TSPO staining intensity of the brain parenchyma was generally weak. Consistent with the reported mitochondrial localization of the TSPO protein, the quality of the staining was punctate [25]. Due to the punctate nature of the cell processes, complete cell shapes could not easily be deciphered. In line with previous observations, the positive immunohistochemical staining for TSPO was cytoplasmatic [57]. Occasionally, positively-stained process-bearing cells were identifiable as glial cells (arrow in Figure 3A), resembling either microglia or astrocytes. Furthermore, TSPO-expressing cells with multiple processes that appeared to be oligodendrocyte progenitor cells were found in the cortex (see Figure 3B). No staining was observed if the primary antibody was omitted (not shown). Additional cells/areas in which TSPO immunoreactivity was observed in the control brains included cekks in the subpial glia (probably both astrocytes and microglia; arrows in Figure 3C), meninges, ependymal cells (arrows in Figure 3D), as well as cells in the choroid plexus (vessels, macrophages, and choroid plexus epithelial cells; see Figure 3E). TSPO^+^ cells, resembling oligodendrocytes, were also observed (arrows in Figure 3F). Again, the staining pattern in these cells tends to be punctate or granular. Furthermore, neuronal cell bodies were labelled with punctuate spots (not shown). The brains of cuprizone-intoxicated mice displayed staining in compartments observed in the control cases (such as in the meninges, neurons and ependymal cells), but also showed markedly enhanced staining in the parenchyma, particularly in regions of demyelination such as the corpus callosum (see Figure 3G–I). In such areas of intense demyelination, ramified glial cells had enhanced TSPO positivity, most of which were clearly identifiable as activated microglia (see Figure 3J/K). TSPO^+^ hypertrophic astrocytes were also present in the brains of cuprizone-intoxicated mice, identified based on their morphology (arrows in Figure 3L).

The densitometrical quantification of TSPO staining intensity revealed significantly higher expression levels in the medial (∆153 ± 15%, *p* < 0.001) and lateral (∆151 ± 11%, *p* < 0.001) corpus callosum, caudoputamen (∆82 ± 15%, *p* < 0.001), thalamus (∆35 ± 6%, *p* < 0.001) and hippocampus (∆128 ± 23%, *p* < 0.05), but not in the cortex region. Due to the discrepancy of cortical TSPO expression based on immunohistochemical evaluation, and in vivo μPET analyses, we additionally quantified the densities of TSPO^+^ cells in the cortex of the control and cuprizone-intoxicated mice, which is a more sensitive method to detect subtle expression differences. As shown in Figure 3N, higher densities of TSPO^+^ cells were found in the cortex of the cuprizone-intoxicated mice compared to the control mice. To verify induced TSPO expression in the cuprizone model, we quantified *Tspo* mRNA expression by rt RT-PCR in the corpus callosum and cortex after 1, 3 and 5 weeks of cuprizone intoxication. As demonstrated in Figure 4, higher *Tspo* expression levels were found in both the white matter corpus callosum and grey matter cortex region. Additionally, the extent of *Tspo* mRNA expression induction was greater in the corpus callosum compared to the cortex.

### 3.3. TSPO Mainly Co-Localizes with Microglia

As stated above, based on their morphology, TSPO^+^ cells were identifiable as glial cells, resembling either microglia or astrocytes. Furthermore, it has been suggested that both activated astrocytes and microglia are responsible for the enhanced TSPO-ligand binding in brain areas undergoing cuprizone-induced demyelination [41,58]. To test for this, IBA1/TSPO and GFAP/TSPO double-labelling experiments were performed, and absolute glial cell densities as well as numbers of glia cells expressing the respective glia cell marker protein and TSPO (IBA1^+^/TSPO^+^ or GFAP^+^/TSPO^+^) was evaluated in the white matter corpus callosum and grey matter cortex after 1, 3 and 5 weeks of cuprizone intoxication. As shown in Figure 5, the densities of the IBA1^+^ cells gradually increased in the corpus callosum during the course of cuprizone-induced demyelination. At week 5, the mean density of the IBA1^+^ cells in the medial corpus callosum was 687.5 ± 300.75 cells/mm^2^. Most of the IBA1^+^ cells also expressed TSPO (665 ± 291.82 cells/mm^2^; ~97%). In line with previous results [59], the extent of microgliosis was less severe and delayed in the cortex (control: 137.5 ± 35.35 cells/mm² versus 5 weeks of cuprizone: 197.5 ± 80.325 cells/mm^2^). Comparable to what we found in the corpus callosum, most of the IBA1^+^ cells in the cortex expressed TSPO at week 5 (192.5 ± 84.2 cells/mm^2^; ~97%). The situation for GFAP expressing cells was different. Although the densities of the GFAP^+^ cells gradually increased in the corpus callosum and cortex during the course of cuprizone-induced demyelination, a significant proportion of the GFAP^+^ cells were TSPO-negative in the control and cuprizone-exposed mice (see Figure 6). At week 5, ~31% of GFAP^+^ astrocytes co-expressed TSPO in the white matter corpus callosum, whereas ~23% of GFAP^+^ astrocytes co-expressed TSPO in the grey matter cortex. To verify TSPO expression in the astrocytes, the brain slides from the cuprizone-treated hGFAP-eGFP-mice were processed for anti-TSPO immunoflourescence staining. These mice express eGFP not only in proximal processes, but also in the fine distal processes of astrocytes [46]. In line with our results that were obtained by fluorescence-double-labelling, the anti-TSPO signal could clearly be localized to the astrocyte cell bodies (see Figure 6C).

### 3.4. TSPO Is Not Expressed in the Mitochondria of Injured Axons

Different groups reported an increase in mitochondrial content (through an increase in size and number) within demyelinated axons in MS as well as following experimental demyelination [53,60,61]. Therefore, increased (18F)-GE180 binding levels might principally also be due to an increase in the axonal mitochondrial load. To test for this possibility, we first used 3D-electron microscopy to observe the accumulation of mitochondria in the corpus callosum of cuprizone-exposed mice. As expected, numerous mitochondria were found in astrocytes (arrowheads in Figure 7A) and myelin-laden microglia cells (arrowheads in Figure 7B). Some mitochondria were also present in normal-appearing, myelinated axons (arrowheads in Figure 7C). Additionally, pronounced mitochondrial accumulation was frequently found at sites of axonal swelling (arrowheads in Figure 7D). 3D reconstruction of serial blockface scanning 3D EM images revealed that in many cases, axonal mitochondrial density was high at the close vicinity of an axonal swelling, but continuously declined with increasing distance to the swelling (data not shown).

To verify the ultrastructural data, the presence of mitochondria in axonal spheroids was additionally visualized by APP/COXIV double-labelling experiments. As demonstrated in Figure 7F, most of the APP^+^ spheroids contained COXIV^+^ mitochondrial profiles. Next, we looked for the co-labelling of APP^+^ axonal spheroids with TSPO. Surprisingly, almost no TSPO^+^/APP^+^ spheroids were found in the corpus callosum of cuprizone-treated mice (Figure 7I). In particular, ~97% of IBA1^+^ microglia cells, ~31% of GFAP^+^ astrocytes, but just ~3% of APP^+^ axonal spheroids co-expressed TSPO, ruling out that increased [18F]-GE180 binding is due to mitochondrial accumulation within axons.

### 3.5. TSPO Is Expressed by Both Microglia and Monocytes in Inflammatory Lesions

The results from most studies suggest that cuprizone-induced pathology is characterized by intense microglia accumulation [62], whereas peripheral immune cells have negligible significance for cuprizone-induced demyelination. This reductionist model therefore does not allow the study of TSPO expression of invading monocytes under demyelinating conditions. As recently published by our group [43,44], the combination of cuprizone intoxication and MOG_35-55_ immunization (i.e., Cup/EAE) results in significant peripheral immune cell recruitment. We took advantage of this model to investigate the expression of TSPO in peripheral immune cells (see Figure 8A for the experimental setup). MOG-EAE and Cup/EAE mice showed clinical signs of disease progression beginning ∼11 days after immunization, with the Cup/EAE-treated mice reaching slightly higher cumulative (9.38 ± 1.14 versus 7.75 ± 1.05) and maximum (3.0 ± 0.2 versus 2.5 ± 0.35) disease scores than MOG-EAE-treated mice (see Figure 8B). In line with our results described above, a significant increase in anti-TSPO staining intensity was observed in the corpus callosum of only cuprizone-treated mice (Figure 8D). In MOG-EAE mice, no significant increase in anti-TSPO staining intensity could be observed in the corpus callosum (Figure 8E). However, in some cases, perivascular foci were in direct vicinity of the third ventricle, and these foci showed pronounced TSPO immunoreactivity (Figure 8F). In Cup/EAE mice, the TSPO staining intensity was pronounced in the cuprizone vulnerable regions, such as the corpus callosum (Figure 8G). A particularly high TSPO staining intensity was observed around perivascular inflammatory lesions (Figure 8H,I). To determine whether microglia or recruited monocytes express TSPO in the Cup/EAE model, we utilized transgenic (CX_3_CR1^+/eGFP^/CCR2^+/RFP^) mice with RFP-expressing monocyte-derived macrophages and eGFP-expressing microglia [47]. As shown in Figure 8J,K, the TSPO signal clearly co-localized to both, eGFP-expressing microglia (arrow) and RFP-expressing monocyte-derived macrophages (arrowhead).

## 4. Discussion

The present study demonstrates the temporal, regional and cellular patterns of TSPO expression during the progression of cuprizone-induced demyelination in the mouse brain. Furthermore, this study indicates a very good applicability for monitoring neuroinflammatory processes using the third-generation TSPO radioligand, (18F)-GE180. To the best of our knowledge, this is the first study investigating cuprizone-intoxicated mice using (18F)-GE180 PET with the aim of quantifying the uptake/binding in various anatomical brain regions, and correlating ligand uptake with TSPO expression.

During the last two decades, the prototypic TSPO radioligand in use has been (11C)-PK11195, with the first human brain studies conducted in patients with Rasmussen’s encephalitis [63], with glioma [64], and in healthy controls and MS patients [32]. Using (11C)-PK11195 results in increased binding in MS lesions, as demonstrated by postmortem studies and in vivo PET [29,65]. Although (11C)-PK11195 showed promise as a biomarker in MS [66], newer TSPO-selective radioligands might be superior to (11C)-PK11195 based on their higher brain uptake and more specific binding [67]. In this study, we were able to demonstrate that cuprizone-induced demyelination is paralleled by an increase in TSPO radioligand (18F)-GE180 uptake. Comparably, increased uptake of different TSPO ligands, namely, [^3^H]-(R)-PK11195 [58,68], (123I)-CLINDE [41], and (125I)iodo-DPA-713 [56], has been shown in the cuprizone model. It was suggested that, in the cuprizone model, the observed increase in [3H]-R-PK11195 binding is a result of an increase in the apparent number of binding sites rather than a change in receptor binding affinity [58]. In line with this assumption, we found higher TSPO expression on the protein (Figure 3) and mRNA (Figure 4) level.

The cellular source of TSPO expression in the cuprizone model is not entirely clear. Mattner and colleagues concluded, based on immunflourescence double-labelling experiments, that one of the main sources of TSPO expression in the demyelinated corpus callosum is activated astrocytes. In these studies, virtually all the GFAP-positive astrocytes as well expressed TSPO [41]. Chen and colleagues performed GFAP or Mac-1 immunohistochemistry combined with (3H)-(R)-PK11195 autoradiography and found co-localization of (3H)-(R)-PK11195 binding with both GFAP-expressing astrocytes and Mac-1-expressing microglia [58]. In the current study, we quantified microglia (anti-IBA1) and astrocyte (anti-GFAP) densities and determined the percentage of these cells expressing TSPO. As shown in Figure 5 and Figure 6, most of the IBA1^+^ cells co-expressed TSPO, whereas just about 30% of the GFAP^+^ cells co-expressed TSPO. Of note, the mapping of cellular expression of proteins in tissue subjected to cuprizone-induced demyelination is not straightforward. After a 5-week cuprizone intoxication protocol, the midline of the corpus callosum is hypercellular, making it difficult to delineate the border of single cells. Furthermore, GFAP is not always expressed within the entire astrocyte cell body but frequently spares one part of its perinuclear compartment. This anti-GFAP staining pattern causes problems in delineating the exact border of a single astrocyte and, therefore, does not always allow to decide whether a certain protein is expressed within an astrocyte or not. Additionally, the staining efficacy of different antibodies or staining protocols might result in more or less co-localization signals. To further verify that astrocytes express TSPO in the cuprizone model, we processed brain sections of hGFAP-eGFP-mice for anti-TSPO immunoflourescence staining. These mice express eGFP not only in the cell body and proximal processes, but also in the fine distal processes of astrocytes [46]. Furthermore, these mice express eGFP not in all but just a minority of astrocytes. This makes the delineation of single astrocytes easy even in regions with high astrocyte density. As demonstrated in Figure 6C, TSPO expression clearly co-localized to the eGFP signal. In summary, our data provide unequivocal evidence that both activated astrocytes and microglia were responsible for the enhanced [18F]-GE180 binding to TSPO in brain areas undergoing demyelination. Moreover, the bulk of TSPO reactivity in the cuprizone model can be attributed to microglia. Notably, histological and imaging findings in different types of MS lesions indicate that activated microglia are inflammatory modulators, including inactive chronic and cortical lesions, as well as in normal appearing white and grey matter areas [7]. Consequently, imaging microglial activation by PET using radiolabeled TSPO ligands should provide a sensitive marker for neuroinflammatory processes.

High TSPO expression has been demonstrated in various macrophage populations such as in synovial macrophages [69], myocardial macrophages [70], or macrophages of atherosclerotic plaques [71]. To determine whether recruited monocytes, in addition to microglia, express TSPO, we combined the toxin-induced demyelination model cuprizone with the autoimmune EAE model (i.e., Cup/EAE). As recently demonstrated by our group, cuprizone-induced oligodendrocyte degeneration triggers peripheral immune cell recruitment into the forebrain [43,44]. Here, we utilized transgenic (CX_3_CR1^+/eGFP^/CCR2^+/RFP^) mice with eGFP-expressing microglia and RFP-expressing monocyte-derived macrophages [47] to investigate which cell type expresses TSPO. As shown in Figure 8J,K, the TSPO signal clearly co-localized to both, eGFP-expressing microglia and RFP-expressing monocyte-derived macrophages. Although not shown in this study, it has been reported that TSPO is as well expressed by CD4^+^ lymphocytes in EAE [72], as it is in humans [73] and in a model of rheumatoid arthritis [69]. Further studies have to demonstrate to what extent lymphocytes and other peripheral immune cells contribute to the increased TSPO ligand uptake observed in MS lesions.

Axonal degeneration is believed to be a major cause of permanent neurological disability in MS patients. It is assumed that myelinated axons contain two populations of mitochondria: stationary and motile mitochondria. It has been shown that the volume of mitochondrial stationary sites is increased in demyelinated CNS axons [53]. In light of these findings, it is of interest to speculate whether some of the white matter TSPO radioligand uptake observed in PET studies is due to mitochondrial binding in acutely injured axons. 3D reconstructions of serial block-face scanning electron microscopy images showed an accumulation of mitochondria within axonal swellings. Furthermore, immunofluorescence double-labelling experiments revealed accumulations of the mitochondrial protein COXIV at sites of amyloid precursor protein (APP) accumulation (see Figure 7). The most commonly-used marker to visualize axonal transport deficits is the APP [74]. APP is an integral glycoprotein type 1, which is transported towards the axon terminal via anterograde axonal transport [75]. In case of a disturbed axonal transport machinery, APP accumulates at the side of axon injury and can then be visualized by immunohistochemistry as so called “spheroids” [76,77]. Although we found a clear co-localization of COXIV and APP, such a co-localization was not found for TSPO and APP. As already stated above, there is growing realization that axonal degeneration plays a pivotal role in the development of neurological disability in patients with MS. In histopathological studies, acute axonal injury, as defined by the accumulation of APP, was found to occur not only in active demyelinating but also in remyelinating and inactive demyelinated MS lesions [78,79]. Although it is currently not clear why mitochondria do not express TSPO within injured axons, diffuse axonal degeneration probably does not contribute to TSPO-ligand uptake in MS tissues.

PET ligands targeting TSPO represent promising tools to visualize neuroinflammation in MS patients. This may provide information beyond contrast-enhanced MRI and is readily applicable to all patients. This study illustrates that the pathological substrate of an increase in TSPO ligand binding might be diverse including microglia activation, peripheral monocyte recruitment, and astrocytosis.

## Figures and Tables

**Figure 1 cells-08-00094-f001:**
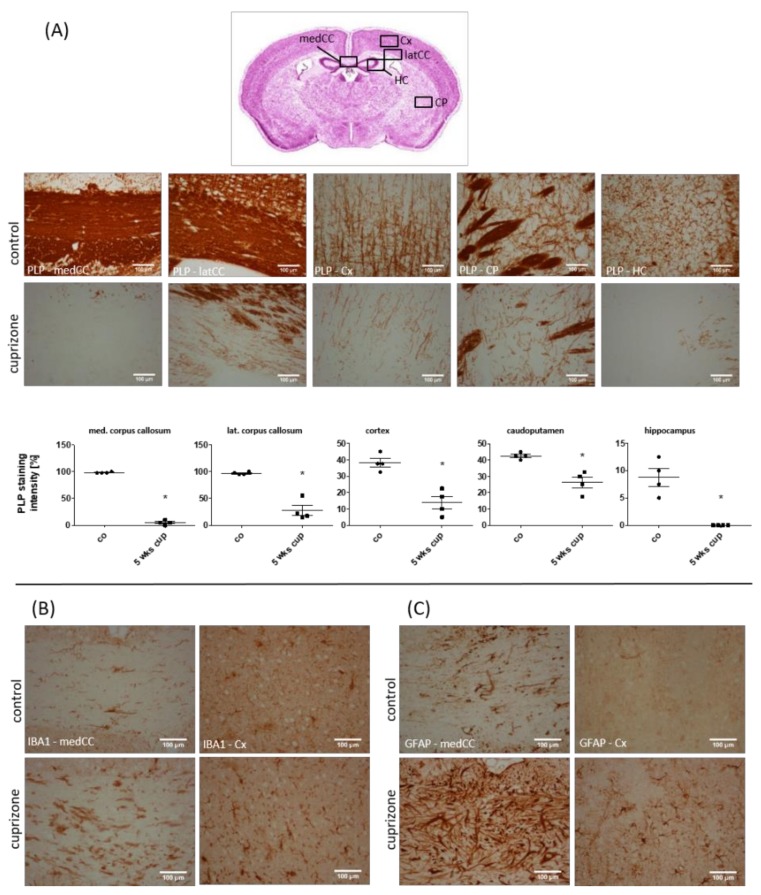
Cuprizone-intoxication induces multifocal demyelination and reactive glia activation. (**A**) The myelination of different brain areas, determined by anti-PLP staining. The upper row shows representative pictures of the control (*n* = 4), and the lower row shows representative pictures of the cuprizone-intoxicated animals (5 weeks; *n* = 4). (**B**) Cuprizone-induced demyelination is paralleled by microglia activation (shown with IBA1 immunoreactivity; *n* = 4), and (**C**) astrocyte activation (shown with GFAP immunoreactivity; *n* = 4). medCC (medial part of the corpus callosum); latCC (lateral part of the corpus callosum); Cx (cortex); CP (caudoputamen); HC (hippocampus). Differences were determined using Mann–Whitney tests. * *p*  <  0.05, ** *p*  <  0.01, and *** *p*  <  0.001.

**Figure 2 cells-08-00094-f002:**
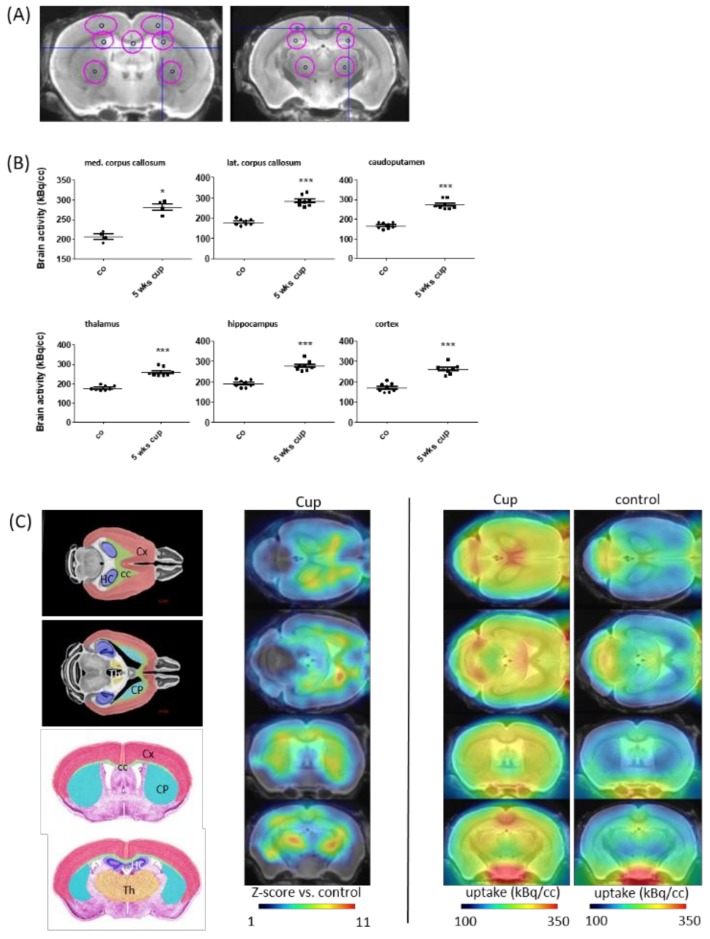
In vivo imaging demonstrates an increase in [18F]-GE180 uptake after cuprizone-intoxication. (**A**) The definition of VOIs (volumes of interest), demonstrated in a mouse brain MRI atlas in the coronal slides: medial corpus callosum (2.2 mm^3^), lateral corpus callosum (2.9 mm^3^), caudoputamen (4.4 mm^3^), thalamus (4.4 mm^3^), hippocampus (4.4 mm^3^), and cortex (6.7 mm^3^). (**B**) The quantification of radioligand uptake in the control animals (*n* = 4) and after a 5-week cuprizone intoxication period (*n* = 4). (**C**) Schematic illustration of the measured areas and cumulative heat map illustrating radioligand uptake in cuprizone-treated mice. The color codes demonstrate the standardized radioligand uptake levels relative to the controls (Z-scores; left column), and the normalized uptake in cuprizone-treated (middle column) and control (right column) mice. Cx (cortex); CC (corpus callosum); HC (hippocampus); Th (thalamus); CP (caudoputamen). Differences were determined using Mann–Whitney tests. * *p*  <  0.05, ** *p*  <  0.01, and *** *p*  <  0.001.

**Figure 3 cells-08-00094-f003:**
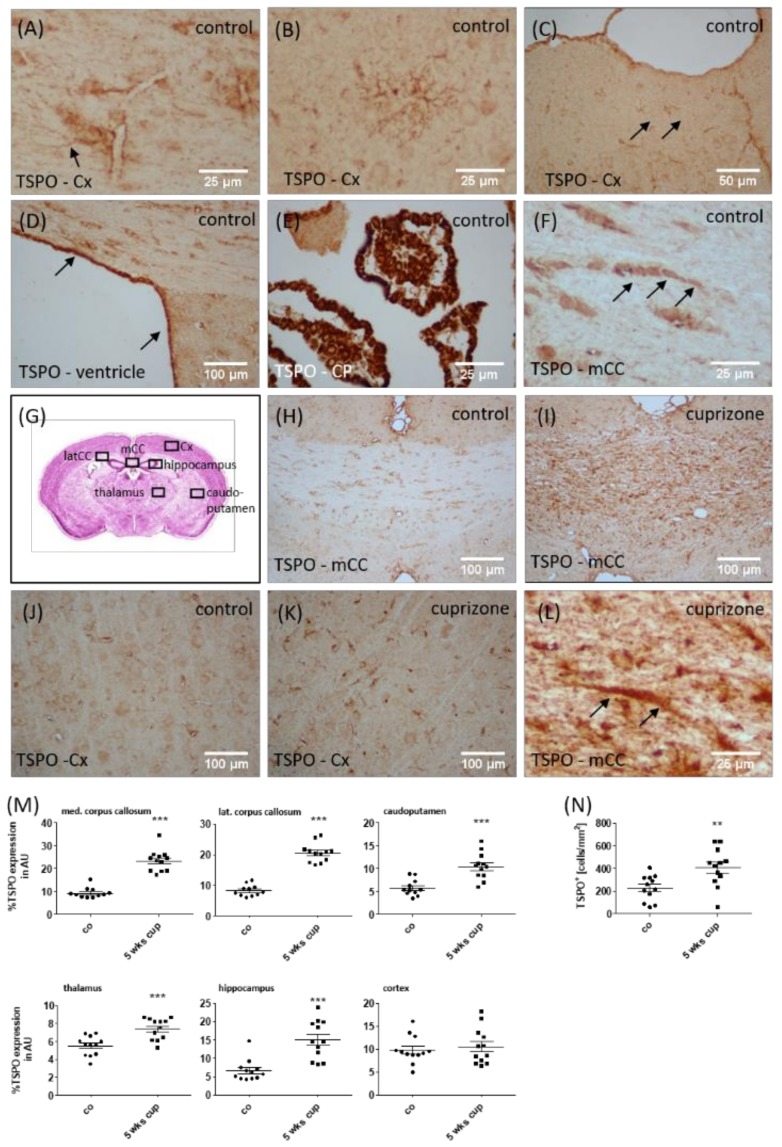
Expression of TSPO in cuprizone-intoxicated mice. The representative results of the anti-TSPO immunohistochemistry demonstrating the expression of TSPO in perivascular astrocytes (arrow in **A**), oligodendrocyte progenitor cells (**B**), glia cells beneath the pial surface of the cortex (**C**), ependymal cells lining the lateral ventricles (**D**), choroid plexus cells (**E**) and interfascicular oligodendrocytes in the corpus callosum (arrows in **F**). (**G**) Schematic illustration of the investigated regions. (**H**–**K**) The representative results of the anti-TSPO immunohistochemistry in the control or cuprizone-intoxicated mice. (**L**) A high-power view of an anti-TSPO^+^ astrocyte (arrows) in the corpus callosum of cuprizone-intoxicated mice. (**M**) The densitometric quantification of the anti-TSPO staining intensity in different brain areas. (**N**) The quantification of anti-TSPO^+^ cell densities in the somatosensory cortex of control and 5-week cuprizone-intoxicated mice, with representative pictures shown in (**J**) and (**K**). mCC (medial part of the corpus callosum); Cx (cortex); CP (choroid plexus). Differences between the groups were statistically tested using two-tailed *t*-test. Welsh correction was applied for the medial corpus callosum. * *p*  <  0.05, ** *p*  <  0.01, and *** *p*  <  0.001.

**Figure 4 cells-08-00094-f004:**
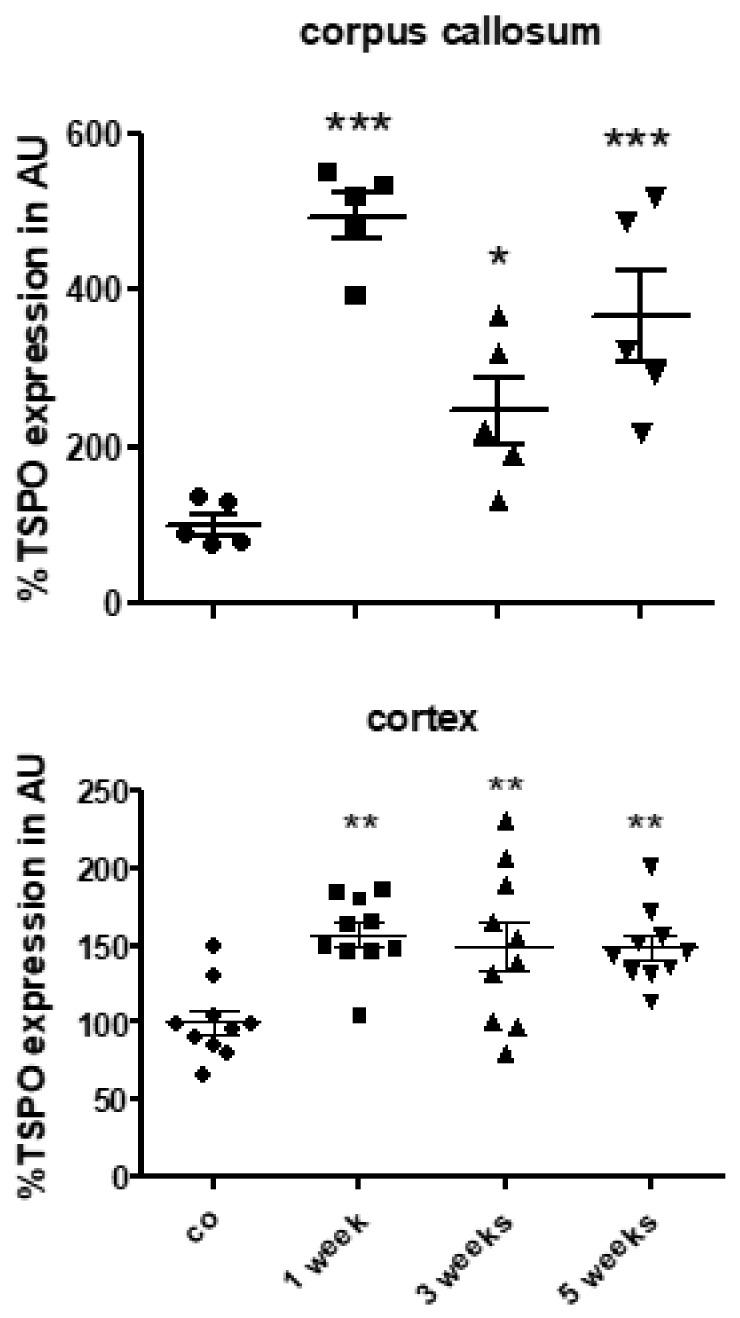
Expression levels of *Tspo* mRNA during the course of cuprizone-induced demyelination. *Tspo* mRNA levels in the control animals (*n* = 5) and the animals intoxicated with cuprizone for one (*n* = 5), three (*n* = 5) or five (*n* = 5) weeks. Differences between the groups were statistically tested using one-way ANOVA with the obtained p-values corrected for multiple testing using the Dunnett’s post-hoc test; * *p* ≤ 0.05, ** *p* ≤ 0.05, *** *p* ≤ 0.001.

**Figure 5 cells-08-00094-f005:**
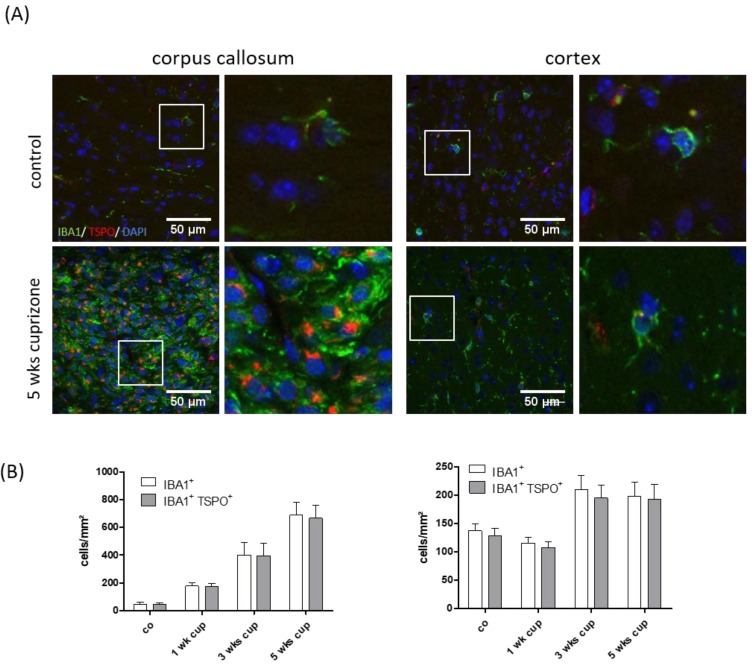
Co-localization of TSPO with the microglia/monocyte marker, IBA1. (**A**) The representative images of IBA1 (green) and TSPO (red) immunofluorescence double stains in the control and 5-week cuprizone-intoxicated mice (left: corpus callosum; right: cortex). The white boxes highlight cells which are shown on the right-hand side at higher magnification. (**B**) The quantification of entire IBA1^+^ cell densities (white columns) and IBA1^+^/TSPO^+^ double positive cell densities (grey columns) in the control animals (*n* = 4) and the animals intoxicated with cuprizone for one (*n* = 5), three (*n* = 5) or five (*n* = 5) weeks.

**Figure 6 cells-08-00094-f006:**
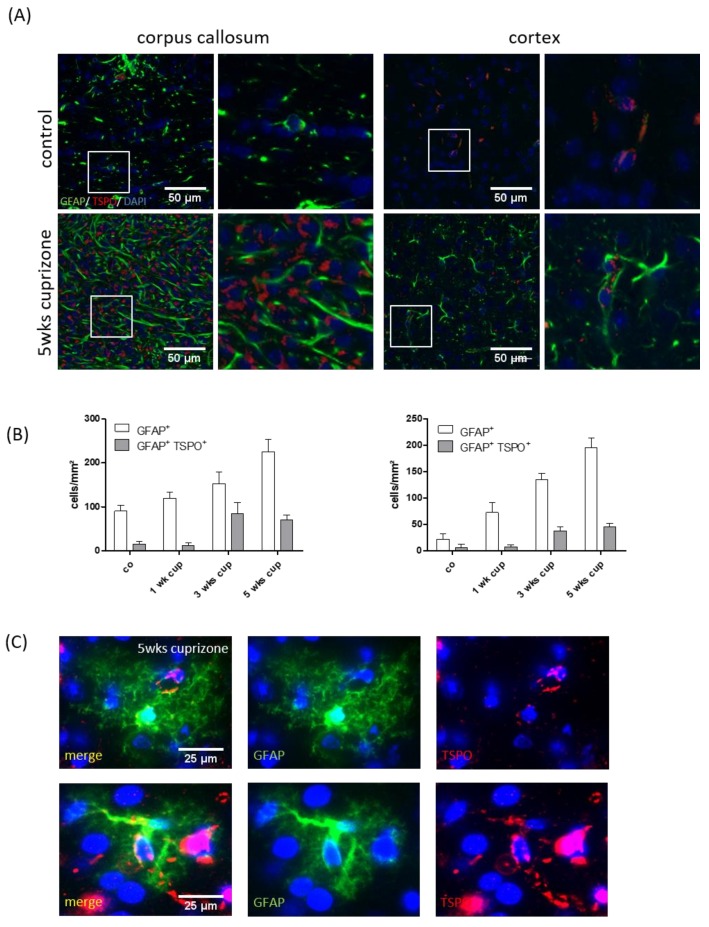
Co-localization of TSPO with the astrocyte marker protein, GFAP (**A**). The representative images of GFAP (green) and TSPO (red) immunofluorescence double stains in the control and 5-week cuprizone-intoxicated mice (left: corpus callosum; right: cortex). The white boxes highlight cells which are shown on the right-hand side at higher magnification. (**B**) The quantification of the entire GFAP^+^ cell densities (white columns) and GFAP^+^/TSPO^+^ double positive cell densities (grey columns) in the control animals (*n* = 4) and the animals intoxicated with cuprizone for one (*n* = 5), three (*n* = 5) or five (*n* = 5) weeks. (**C**) The representative images demonstrating expression of TSPO (red) in eGFP-expressing astrocytes (green).

**Figure 7 cells-08-00094-f007:**
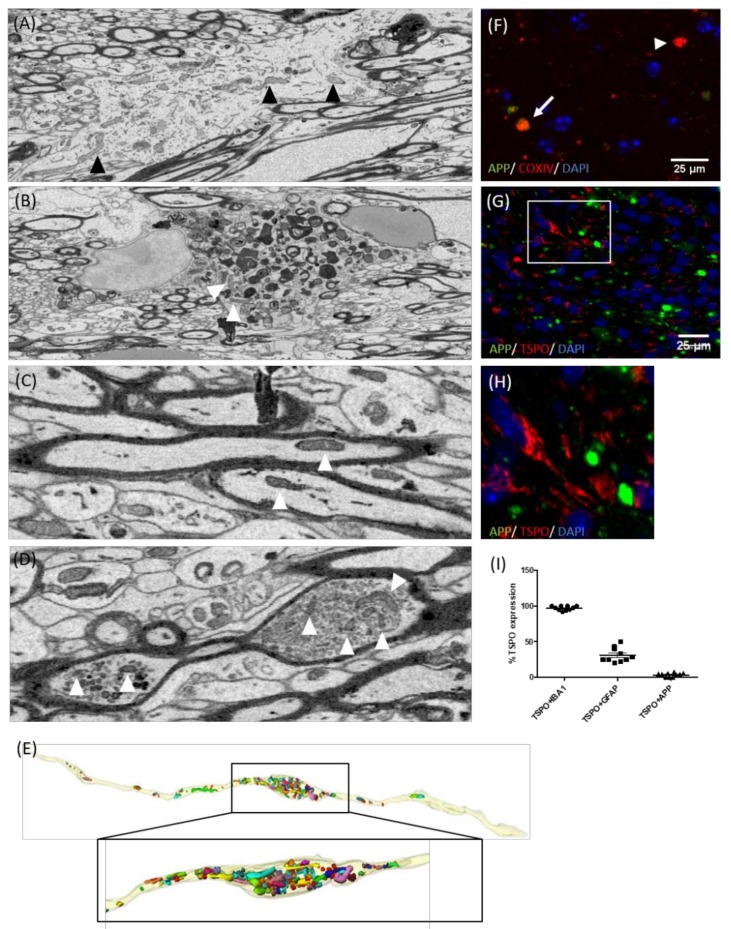
Accumulation of mitochondria in the damaged axons. The representative electron microscopic images demonstrating mitochondria (arrowheads) in the astrocytes (**A**), phagocytosing microglia (**B**), myelinated axons (**C**), or axonal swellings (**D**). The mice were intoxicated with cuprizone for three weeks (*n* = 4). The arrowheads highlight mitochondria. (**E**) 3D reconstruction of an axonal swelling. The mitochondria are highlighted by different colors. (**F**) The representative image of APP (green) and COXIV (red) immunofluorescence double stains in the 5-week cuprizone-intoxicated mice. The arrowhead highlights a COXIV^+^ single-positive, and the arrow highlights a COXIV^+^/APP^+^ double-positive spheroid. (**G**) The representative image of APP (green) and TSPO (red) immunofluorescence double stains in the 5-week cuprizone-intoxicated mice. The white box highlights a region which is shown in (**H**), at higher magnification. (**I**) The percentage of microglia (IBA1^+^), astrocytes (GFAP^+^) or axonal spheroids (APP^+^) co-expressing TSPO.

**Figure 8 cells-08-00094-f008:**
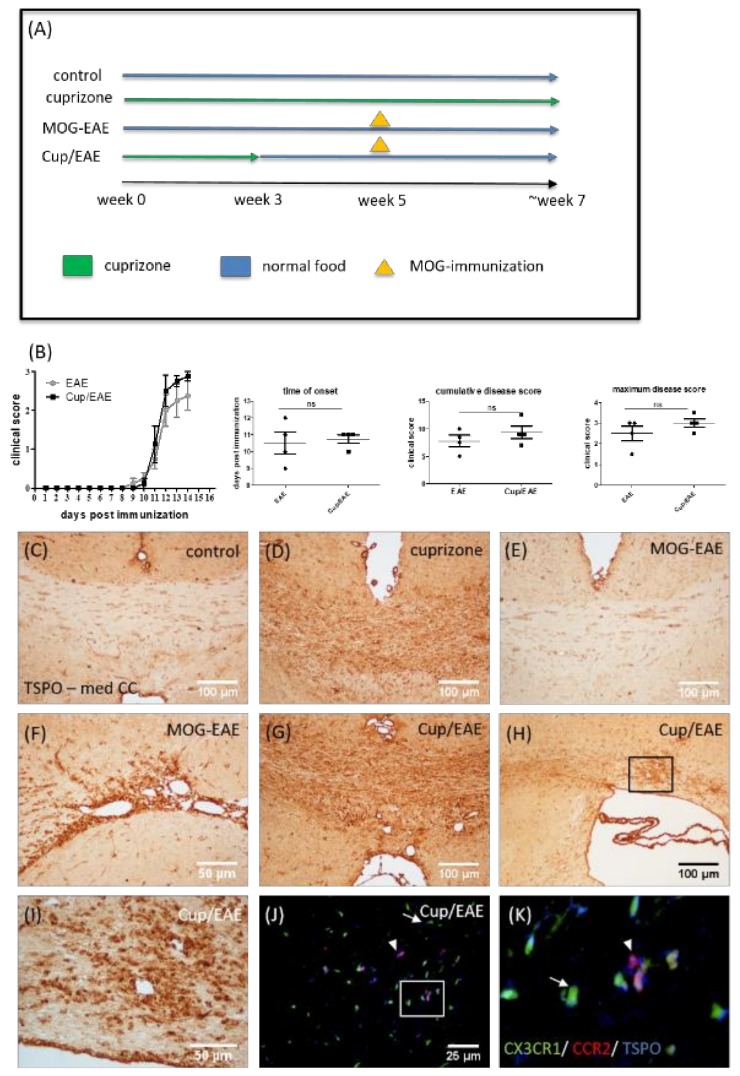
Expression of TSPO in Cup/EAE mice (**A**). Schematic depicting the experimental setup. The periods of normal chow diet are highlighted in blue. The periods of cuprizone intoxication are highlighted in green, and the time point of MOG_35–55_ immunization with subsequent active EAE development are highlighted in orange. (**B**) The clinical scores for MOG-EAE and Cup/EAE treated mice with the time of onset, cumulative disease score and maximum disease score (*n* = 4). The representative images of anti-TSPO immunohistochemistry in the controls (**C**), cuprizone-intoxicated (**D**), MOG-EAE (**E**,**F**), and Cup/EAE (**G**–**I**) mice. The perivascular lesion, highlighted in (**H**) by the black box, is shown in (**I**) at higher magnification. (**J**,**K**) The representative images of the corpus callosum from cuprizone-treated mice, showing microglia (green), infiltrating monocytes (red) and TSPO (blue). The arrow indicates TSPO expression in microglia cells, and the arrowhead indicates TSPO expression in invaded monocytes.

**Table 1 cells-08-00094-t001:** List of primers used in this study (AT—annealing temperature; bp—base pair length)**.**

	Sense	Antisense	Bp	AT
TSPO1	GCCTACTTTGTACGTGGCGAG	CCTCCCAGCTCTTTCCAGAC	152	60
β-Actin 1	GTACCACCATGTACCCAGGC	AACGCAGCTCAGTAACAGTCC	247	60

**Table 2 cells-08-00094-t002:** List of the antibodies used in this study.

Antigen	Species	Dilution	HIER Method	Purchase Number	RRID	Company
**Single stains**						
PLP	Mouse	1:5,000	None	MCA839G	AB_2237198	Bio-Rad Laboratories, Inc., Germany
TSPO/PBR	Goat	1:100	Tris/EDTA	ab118913	AB_10898989	Abcam, UK
Iba1	Rabbit	1:5,000	Tris/EDTA	019–19741	AB_839504	Wako, USA
GFAP	Chicken	1:5,000	Citrate	ab4674	AB_304558	Abcam, UK
**Double stains**						
TSPO/PBR	Goat	1:100	Tris/EDTA	ab118913	AB_10898989	Abcam, UK
GFAP	Mouse	1:400	Tris/EDTA	G3893	AB_477010	Sigma-Aldrich, USA
Iba1	Rabbit	1:2,500	Tris/EDTA	019-19741	AB_839504	Wako, USA
APP	Mouse	1:1,000	Tris/EDTA	MAB348	AB_94882	Millipore, Germany
COXIV	Rabbit	1:500	Tris/EDTA	ab16056	AB_443304	Abcam, UK

**Table 3 cells-08-00094-t003:** List of the secondary antibodies used in this study.

	Order Number	RRID	Supplier
Rabbit anti-goat IgG	BA-5000	AB_2336126	Vector Laboratories, Burlingame, USA
Goat anti-rabbit IgG	BA-1000	AB_2313606	Vector Laboratories, Burlingame, USA
Goat anti-mouse IgG	BA-9200	AB_2336171	Vector Laboratories, Burlingame, USA
Goat anti-chicken IgG	BA-9010	AB_2336114	Vector Laboratories, Burlingame, USA
Cy3 donkey anti-goat IgG	705-165-147	AB_2307351	Jackson ImmunoResearch Labs
Alexa Fluor 488 donkey anti-rabbit IgG	A21206	AB_2535792	Invitrogen, USA
Alexa Fluor 488 donkey anti-mouse IgG	A21202	AB_141607	Invitrogen, USA

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
