# Peer review of "Expression of Translocator Protein and [18F]-GE180 Ligand Uptake in Multiple Sclerosis Animal Models"

_cells, 2019, doi:10.3390/cells8020094_

Round 1

Reviewer 1 Report

The manuscript entitled „Expression of Translocator Protein and [18F]-GE180 Ligand Uptake in Multiple Sclerosis Animal Models (Nack et al.) represents an interesting and well-designed study, which characterized the distribution of the mitochondrial translocator protein (TSPO) in the brain of mice and underlines that positron emission tomography (PET) and [18F]-GE180, as a ligand of TSPO may be used for visualization and quantification of neuroinflammation and neurodegeneration.

The data were obtained in animal models that imitate the pathogenesis of different types of multiple sclerosis (MS) and induce toxic or autoimmune lesions in the brain (cuprizone-induced demyelination and MOG-induced autoimmune encephalomyelitis, respectively).  Using the immunohistochemical, immunofluorescent, real-time reverse transcription-PCR analyses the authors clearly showed that Cup intoxication was followed by enhanced expression of TCPO in microglial cells and hypertrophic astrocytes in corpus callosum, cortex, hippocampus, thalamus and caudoputamen. Besides, in combined Cup/EAE model they visualized the presence of TCPO around perivascular inflammatory lesions, as well as in monocyte-derived macrophages and microglia of transgenic (CX3CR1+/eGFP/CCR2+/RFP) mice. Moreover, by 3D-electron microscopy they also showed that in affected cells and in sites of focal axonal injury accumulated mitochondria. However, according to APP/COXIV double labelling, these mitochondria did not express TSPO. 

Importantly, the authors also showed that in Cup-fed mice the areas of demyelination and glial activation were owing to the TSPO hiperexpression detectable also by micro-PET imaging with [18F]-GE180, indicating that this the TSPO tracer might be used also in MS patients, as a valuable marker for the in vivo visualization of subtle neuropathological changes in the brain.

Owing to the novelty of the results and well documented state of the art in the discussion I would recommend the fast publication of this interesting study.

Minor objections are the following:

Line 114        Replace number 8 at the beginning of the sentence with word „eight“

Line 178        Correct the tip-feller PBS perfusiontissues

Figure 1.      A) In representative coronal section of the brain the abbreviated names of areas            selected for analyses might be also shown

Author Response

#1 Reviewer

Thank you very much for taking the time and effort to review our joint manuscript.

Q1: Line 114-Replace number 8 at the beginning of the sentence with word „eight“

A1: Thank you for this notion. We have changed this part in the revised version of the manuscript.

Q2: Line 178- Correct the tip-feller PBS perfusion tissues

A2: Thank you for this notion. We have changed this part in the revised version of the manuscript.

Q3: Figure 1-A) In representative coronal section of the brain the abbreviated names of areas selected for analyses might be also shown

A3: Thank you for this notion. We have changed the figure in the revised version of the manuscript. The abbreviations are now given in the figure legend of the figure 1.

Reviewer 2 Report

The authors present ineresting article concerning  expression of Translocator protein and [18F]-GE180 ligand uptake in multiple sclerosis animal models. The materials and methods are correctly described. The animal  experiments were approved by bioethical commitee. The results are discussed correctly. However, the discussion could longer. Especially, lack of expression in mitochondria could be more extensively descrbed.

Author Response

#2 Reviewer

Thank you very much for taking the time and effort to review our joint manuscript.

Reviewer 3 Report

Nack et al. provide in their manuscript an in depth analysis of a third generation tracer of translocator protein (TSPO) in positron emission tomography (PET). The paper is nicely written and includes high quality data using this tracer in different mouse models of Multiple Sclerosis.

As the authors state, [18F]-GE180 has been already used in PET analysis on people with relapsing remitting MS (Vomacka et al, EJNMMI Res 2017). The here presented data aims to investigate this technique in more detail using different mouse models, and to clarify the cell types expressing TSPO in this circumstance.

Introduction:

In the introduction the authors give a nice overview on MS, TSPO and the need for alternative measures. However, MS has a strong autoimmune component and this aspect is a bit overlooked. Not only monocytes but for example B and T cells enter the central nervous system and are strongly involved in the pathogenesis. These cells express TSPO as well (as acknowledge by the authors in the discussion) and reasoning has to be given why they are not analysed here. In particular if the aim of the paper is to " readdress which cell types express TSPO after experimental demyelination". 

In general, the aim of the study, the rational and the potential benefits should be worked out a bit better.

Methods:

The methods are clearly written, very extensive and to the best of my judgement complete.

Results:

The results are mostly clearly displayed (see general comments below) and of high quality.

Interestingly, the authors state that in MOG-EAE no increase in TSPO staining was observed. While this seems to be right for Fig 8D I am not sure if you can say this about figure 8E. In this line, did these mice develop signs of disease? What was the EAE score of the mice displayed? What timepoint of the disease was analysed? Can you quantify these claims? How many mice have been analysed? Does that result mean infiltrating monocytes in MOG-EAE do not express TSPO and/or do not activate microglia? Please provide details on these experiments (scores, timepoints...) and discuss.

Discussion

Similar to the introduction, the novelty, relevance and rational for this study could be worked out a bit better.

General comments and suggestions:

Starting from the introduction, references seem to be off by one - please double check.

It would help the readability of the paper to label or number the areas analysed in the overview brain sections.

It would further be important to include scale bars within all histology pictures.

It further would be important to show some control images. How does a non treated mouse brain look with the imaging technique for example in figure 2?

To improve the quality of the quantification in histograms it would be important to display the single measures within the histograms.

Overall, I think this is a very solid paper with clean data and would highly support publication after these suggestions are addressed.

Author Response

#3 Reviewer

Q1: Nack et al. provide in their manuscript an in depth analysis of a third generation tracer of translocator protein (TSPO) in positron emission tomography (PET). The paper is nicely written and includes high quality data using this tracer in different mouse models of Multiple Sclerosis. As the authors state, [18F]-GE180 has been already used in PET analysis on people with relapsing remitting MS (Vomacka et al, EJNMMI Res 2017). The here presented data aims to investigate this technique in more detail using different mouse models, and to clarify the cell types expressing TSPO in this circumstance. In the introduction the authors give a nice overview on MS, TSPO and the need for alternative measures. However, MS has a strong autoimmune component and this aspect is a bit overlooked. Not only monocytes but for example B and T cells enter the central nervous system and are strongly involved in the pathogenesis. These cells express TSPO as well (as acknowledge by the authors in the discussion) and reasoning has to be given why they are not analyzed here. In particular if the aim of the paper is to "readdress which cell types express TSPO after experimental demyelination". 

A1: Thank you for this comment. We currently work on another project addressing the functional role of TSPO. There we plan to use TspoloxP mice. These floxed mutant mice possess loxP sites flanking exons 2 and 3 of the Tspo gene. Expression of TSPO in different peripheral immune cell populations will be addressed by flow cytometry along with in situ hybridization and immunohistochemistry. We are positive that we can publish these descriptive and mechanistic results soon and hope that the reviewer(s) will follow our argumentation that these results are not shown in this manuscript. Nevertheless, the discussion now includes more relevant citations showing that TSPO is expressed in peripheral immune cells, including lymphocytes.

Q2: In general, the aim of the study, the rational and the potential benefits should be worked out a bit better.

A2: Thank you for this notion. We have changed this part in the revised version of the manuscript.

Q3: Interestingly, the authors state that in MOG-EAE no increase in TSPO staining was observed. While this seems to be right for Fig 8D I am not sure if you can say this about figure 8E.

A3: Thank you for this comment. Indeed we have higher TSPO expression levels in MOG-EAE at sites of periventricular inflammation, which is commonly seen in MOG-EAE mice. We are sorry that we have not made this point clear and now state “In MOG-EAE mice, no significant increase in anti-TSPO staining intensity could be observed in the corpus callosum (figure 8D). However, in some cases, perivascular foci were in direct vicinity of the 3rd ventricle, and these foci showed pronounced TSPO immunoreactivity (figure 8E).”

Q4: In this line, did these mice develop signs of disease? What was the EAE score of the mice displayed? What timepoint of the disease was analysed? Can you quantify these claims? How many mice have been analysed? Does that result mean infiltrating monocytes in MOG-EAE do not express TSPO and/or do not activate microglia? Please provide details on these experiments (scores, timepoints...) and discuss.

A4: Thank you for this comment. Indeed, these mice did develop signs of clinical disease and the requested information are now given in the revised version of the manuscript.

Q5: Similar to the introduction, the novelty, relevance and rational for this study could be worked out a bit better.

A5: Thank you for this notion. We have changed this part in the revised version of the manuscript.  

Q6: Starting from the introduction, references seem to be off by one - please double check.

A6: Thank you for this notion. We have changed this part in the revised version of the manuscript.

Q7: It would help the readability of the paper to label or number the areas analysed in the overview brain sections.

A7: Thank you for this notion. We have changed this part in the revised version of the manuscript.

Q8: It would further be important to include scale bars within all histology pictures.

A8: Thank you for this suggestion. The scale bars have been included in the revised version of the manuscript.

Q9: It further would be important to show some control images. How does a non treated mouse brain look with the imaging technique for example in figure 2?

A9: Thank you for this comment. The results in figure 2C are shown as cumulative heat map, accumulating the treatment (i.e. cuprizone) versus the control measurements. Therefore, no control images are shown. We should point out that regional ligand binding rates were assessed by voxel-wise PET analyses, and were co-registered to an MRI mouse brain atlas (Dorr et al., 2007). This information is now given in the revised version of the manuscript.  

Q10: To improve the quality of the quantification in histograms it would be important to display the single measures within the histograms.

A10: Thank you for this comment. We have performed the suggested changes.

Round 2

Reviewer 3 Report

The authors addressed all my suggestions concerning the figures appropriately. I believe this clearly enhanced the quality of the data presentation. However, it is not clear to me, how the authors addressed my concerns around the text. They state, that they did address it but I don't see these changes. Comparing documents did not clarify any clear chances to increase the clarity around the suggested points. To finish my review I would like to see the new manuscript with track changes or at least the improved bits highlighted.

Author Response

Thank you for this comment. Indeed we missed to include these changes in the revised version of the manuscript. We have now highlighted the respective sections in yellow (see introduction). Another part of the discussion section is as well highlighted in yellow and shows a statement addressing expression of TSPO in lymphocytes. This part was already included in the revised version of the manuscript.